# The Impact of a Diet Containing Sucrose and Systematically Repeated Starvation on the Oxidative Status of the Uterus and Ovary of Rats

**DOI:** 10.3390/nu11071544

**Published:** 2019-07-08

**Authors:** Joanna Sadowska, Wioleta Dudzińska, Ewa Skotnicka, Katarzyna Sielatycka, Izabela Daniel

**Affiliations:** 1Department of Human Nutrition Physiology, Faculty of Food Sciences and Fisheries, West Pomeranian University of Technology, ul. Papieża Pawła VI 3, 71-459 Szczecin, Poland; 2Department of Functional Diagnostics and Physical Medicine, Faculty of Health Sciences, Pomeranian Medical University in Szczecin, ul. Żołnierska 54, 71-210 Szczecin, Poland; 3Department of Physiology, Faculty of Biology, University of Szczecin, ul. Felczaka 3c, 71-412 Szczecin, Poland

**Keywords:** sucrose, starvation, carbohydrate-lipid metabolism, oxidative status, uterus, ovary, rat

## Abstract

The effect of a sucrose diet and repeated one-day starvation on oxidative status in the ovary and uterus is still unknown. Our analysis focused on carbohydrate-lipid metabolism parameters and the changes in red blood cells, ovary and uterus superoxide dismutase (SOD), catalase (CAT), glutathione peroxidase (GPx) activities and malonylodialdehyde (MDA) concentration in rats fed with a diet containing 16% of sucrose and subjected to systematic one-day starvation when using such a diet. It was found that a diet with 16% sucrose contributed to the increase of antioxidant enzyme activity in the blood (GPx and CAT) and uterus (SOD), without changes in MDA concentrations, which indicates an increase in reactive oxygen species (ROS) concentration in these tissues, being balanced by an increase in antioxidant enzyme activity. The introduction of a regular one-day starvation period into the diet intensified oxidative stress and led to a redox imbalance in the reproductive tissues of female rats. This was manifested by higher GPx activity, lower CAT activity and higher MDA concentration in the uterus and lower GPx and CAT activities and lower MDA concentration in the ovaries. The observed changes may be the cause of fertility disorders and possible problems with fertilised egg cell implantation into the uterine tissue.

## 1. Introduction

The role of excessive sugar intake on health and disease is currently an active area of scientific and political debate. The World Health Organization (WHO) strongly recommends adults and children reduce their daily intake of free sugars to less than 10% of their total energy intake. According to a conditional recommendation, a further reduction of free sugar intake to a level below 5% of total energy would provide additional health benefits [1]. Despite these recommendations, sugar consumption remains high in both developed and developing countries [2]. The available data suggest that in Europe the percentage of sugars in total energy were highest in infants (<4 years), with mean values ranging from 20.0% to 38.4%, and decreased over the lifespan to 13.5–24.6% in adults [3]. The data collected by Newens and Walton [4] across the world also suggest that sugars (all mono-and disaccharides) as a percentage of energy are highest in infants and young children, with an average value of 28%, and decrease over the lifespan to 20% in adults.

Since chronic intake of a sugar-rich diet increases overall energy intake and body weight, some young women follow a regularly repeated one-day starvation diet for “body cleansing” and weight maintenance or reduction [5].

It has been shown that one of the factors facilitating the initiation and progression of metabolic disorders in the course of chronic high-sucrose consumption may be a disturbance of prooxidative and antioxidative balance [6].

There are numerous reports on the increased oxidative damage to human or experimental animal plasma, liver, skeletal muscles and heart in the course of a high-sucrose diet. The effect of a sucrose diet on oxidative stress in ovary and uterus is still unknown. However, it is well known that a balance between reactive oxygen species (ROS), such as superoxide anion radical and hydrogen peroxide and antioxidants components, such as catalase (CAT), superoxide dismutase (SOD), glutathione peroxidase (GPx), is essential to enhance steroidogenesis, ovulation and implantation [7]. ROS play a crucial role in follicle rupture induced by gonadotropins [8]. The main sources of physiological ROS production in steroidogenic tissues are mitochondria and metabolic processes [9], but excessive ROS can affect a tissue by generating excess lipid peroxidation, oxidation, and different lesions in DNA [10]. When there are deviations from their normal physiological ranges, it may lead to several diseases in these tissues [11]. Oxidative stress participates in the pathogenesis of endometriosis, unexplained infertility, anovulation, and the impairment of oocyte quality in the reproductive system. Therefore, there is a delicate balance between ROS and antioxidant species the ovary and uterus.

Since reproduction and metabolism in females are closely related and reciprocally regulated [12,13], it has been hypothesised that a sucrose diet and a starvation diet are the cause of carbohydrate-lipid metabolism disturbances and oxidative stress in tissues such as the ovaries or uterus. As regards the prooxidant-antioxidant balance, our analysis focused on the changes in red blood cells, ovary and uterus superoxide dismutase (Cu/Zn-SOD, EC 1.15.1.1), catalase (CAT, EC 1.11.1.6), glutathione peroxidase (GPx, EC 1.11.1.9) activities and malonylodialdehyde (MDA) concentration.

This study aimed at investigating the effect of sucrose administered in the amount of 16% of diet composition (18% of total energy) and a one-day starvation diet repeated once a week on the carbohydrate-lipid metabolism and free radicals processes in the ovaries and uterus of female rats. The sugar amount was set, taking into account the composition of modern diets of many people in which the percentage of added sugars is up to 20% [14].

## 2. Materials and Methods

### 2.1. Animals and Study Design

The study was conducted on 30 female Wistar rats, 3 months old, with the initial body weight of 204.7 ± 15.1 g. The animals were purchased from the Department of Toxicology of Poznań University of Medical Sciences. After a week of adaptation (drinking water and basic feed) to new vivarium conditions (temp. 21–22 °C, relative humidity 55–60%, light-dark cycle 12/12 h), the animals were divided into three groups (*n* = 10 each) of equal body weight, housed in individual cages, fed ad libitum pelleted feeds. All the protocols were approved by the Local Research Ethics Committee (Approval No. 18/2015) in accordance with the European Convention for the Protection of Vertebrate Animals used for Experimental and other Scientific Purposes, Council of Europe, Strasbourg 1986.

The basic feed corresponded to the nutrient requirements AIN-93M [15] and mainly included whole wheat and maize grain. In the modified feed, part of wheat grain and maize grain from the basic feed was replaced by wheat flour (type 500) and sucrose. The fodders were designed to reflect the changes taking place nowadays in the composition of diets which contain simple sugars and refined carbohydrates. The detailed composition of fodders used in the experiment are presented in Appendix A.

The prepared feed mixes were subjected to chemical analysis [16] in order to determine the content of total nitrogen (Kjeldahl method, using a Kjeltec 2100 system from Foss Tecator Hillerod, Denmark) converted into an amount of protein; crude fat (Soxhlet method, using a Soxtec HT6 system from Foss Tecator Hillerod Denmark), crude fiber with the gravimetric method (ANKOM 220 Fiber Analyzer, ANKOM Technology, Macedon, NY, USA), dry matter and ash (gravimetric method, using SUP-4M laboratory dryer Wawa-Med, Warsaw, Poland and muffle furnace Czylok Jastrzębie Zdrój, Poland). The content of carbohydrates was estimated from the difference between dry mass and the sum of the other solid ingredients. The content of gross energy and metabolic energy was estimated using the commonly applied energy equivalents [17]. The content of zinc, cooper, selenium and iron was also determined in feeds by atomic absorption spectrophotometry on ICE-3300 (Thermo Scientific, Walthom, MA, USA).

During the experiment, group I (CG—control group) received the basic feed, group II (SG—sugar group) received the modified feed, over the whole experimental period, and group III (PSG—periodically starved group) received the modified feed and was starved one day in every week (6 days on the modified feed + 1 day starvation). Animals from all groups received tap water to drink.

The experiment lasted 8 weeks, during which the amounts of consumed feed were evaluated. After the eighth week of treatment, we started to sacrifice rats and collect samples after determining that the rats were in the oestrus phase. The status of the oestrus phase was determined by examining the type and abundance of cells present in the vaginal lavage, according to the methodology described by Marcondes et al. [18]. Body weight was also recorded at this time. The assigned rats were fasted overnight (12 h), and anaesthetised with an intramuscular injection (10 mg/kg b.w.) of Ketanest (Pfizer Ireland Pharmaceuticals, Ireland).

### 2.2. Sample Collection

Blood was taken from the heart by cardiac puncture and collected into vacuum tubes (Sarstedt, Germany) with EDTA anticoagulant for erythrocyte and plasma analysis.

Immediately after blood collection, the ovaries and uterus were dissected and weighed to the nearest 0.001 g. The tissues were washed with phosphate-buffered saline (PBS) solution, pH 7.4, immediately frozen in liquid nitrogen and stored at −70 °C until analysis.

Blood samples were immediately centrifuged (1000× *g*, 10 min, 4 °C) to separate plasma and erythrocytes.

Erythrocytes were washed three times with cold phosphate-buffered NaCl solution (phosphate saline buffer: 0.01 M phosphate buffer, 0.14 M NaCl, pH 7.4), chilled to 4 °C, and finally frozen at −70 °C. Plasma was divided into aliquots and immediately deep-frozen at −70 °C until analysis, but not longer than one month.

### 2.3. Biochemical Analysis

Plasma glucose (Trinder’s glucose oxidase method, BioSystems S.A., Cat. No. 11503), triglycerides (glycerol-3-phosphate oxidase method, BioSystems S.A., Cat. No. 11828), total cholesterol (cholesterol oxidase method, BioSystems S.A., Cat. No. 11805), HDL-cholesterol (direct method, BioSystems S.A., Cat. No. 11557), LDL-cholesterol (direct method, BioSystems S.A., Cat. No. 11585) concentration were determined on the Metertech SP-8001 spectrophotometer (Metertech, Taipei, Taiwan) by the enzymatic colorimetric method.

Insulin (Rat ELISA kit Demeditec Diagnostics, Kiel, Germany, Cat. No. DE2048) end estradiol (Rat ELISA kit Fine Test Wuhan Fine Biotech Co., Wuhan, China, Cat. No. ER1507) were assayed using a monoclonal antibody against rat insulin/estradiol, according to the manufacturer’s instructions, on an EnVision apparatus (PerkinElmer Inc., Waltham, MA, USA).

The index of homeostasis model assessment of insulin resistance (HOMA-IR) was calculated as fasting plasma glucose [mM] and fasting plasma insulin [mU/L] divided by 22.5.

The concentration of malonyldialdehyde (MDA) as a product of lipid peroxidation was assayed in blood plasma and in ovarian and uterine homogenate supernatants.

Superoxide dismutase (SOD), catalase (CAT) and glutathione peroxidase (GPx) activities were determined in a red blood cell lysate and in ovarian and uterine homogenate supernatants.

Manufacturers and catalog numbers of reagents used in biochemical tests are presented in Appendix A.

### 2.4. Blood Analysis

Before the analysis, erythrocyte samples were hemolysed by adding a fourfold volume of ice-cold high-performance liquid chromatography (HPLC)-grade water. The hemolysed erythrocyte samples were centrifuged at 10,000× *g* for 15 min at 4 °C, and the supernatant was collected, in which CAT, SOD and GPx activities were determined.

The haemoglobin concentration of hemolysates was determined using the Drabkin’s method to express the enzyme activities per gram of haemoglobin.

Blood glutathione peroxidase analysis: GPx enzyme activity was measured with a GPx Assay Kit (Cayman Chemical, Ann Arbor, MI, USA, Cat. No. 703102), which is commercially available. It was used in accordance with the manufacturer’s protocol. Red blood cell (RBC) samples were diluted to 1:10 using a sample buffer. A coupled reaction with glutathione reductase was used to measure the GPx activity indirectly. The reactions were initiated by adding cumene hydroperoxide. In order to obtain five time points, the absorbance had to be read every minute at 340 nm for 5 min using a plate reader, where one unit was defined as the amount of enzyme that would cause the oxidation of 1.0 nM of NADPH to NADP+ per minute at 25 °C, the GPx activity was expressed as unit per millilitre.

Blood superoxide dismutase analysis: SOD enzyme activity was measured using an SOD Assay Kit (Cayman Chemical, Ann Arbor, MI, USA, Cat. No. 706002), which is commercially available. It was used in accordance with the manufacturer’s protocol. RBC samples were diluted to 1:100 using a sample buffer. The measurement was based on the activity of SOD in the sample in order to cause dismutation of the superoxide radicals generated by xanthine oxidase and hypoxanthine. The absorbance was read at 450 nm. One unit of SOD was defined as the amount of enzyme needed to exhibit 50% dismutation of the superoxide radicals.

Blood catalase analysis: CAT enzyme activity was measured using a CAT Assay Kit (Cayman Chemical, Ann Arbor, MI, USA; Cat. No. 707002), which is commercially available. It was used in accordance with the manufacturer’s protocol. The peroxidatic function of the enzyme was utilised by the assay. Formaldehyde served as the standard, and samples were diluted with a sample buffer to 1:1000 for RBC prior to the assay. H_2_O_2_ was added to initiate the reaction. The assay is based on CAT reactions with methanol in the presence of H_2_O_2_. Where one unit was defined as the amount of enzyme that would cause the formation of 1.0 nM of formaldehyde per minute at 25 °C, the absorbance was read at 540 nm with a use of a plate reader.

Plasma malondialdehyde analysis: MDA concentrations were determined using ELISA kits from EIAab (Wuhan EIAab Sciences Co., Ltd., Wuhan, China; Cat. No. E0597Ge), which is commercially available. It was used in accordance with the manufacturer’s protocol. A quantitative sandwich enzyme immunoassay technique was applied in addition to the main MDA ELISA kit. A monoclonal antibody specific for MDA was used to pre-coat the microtiter plate. Then, calibrators or samples were added to the microtiter plate wells and if MDA was present, it would bind to the antibody pre-coated wells. In order to quantitatively determine the amount of MDA present in the sample, a standardised preparation of horseradish peroxidase (HRP)-conjugated polyclonal antibody, specific for MDA, was added to each well to sandwich the MDA immobilised on the plate. The microtiter plate was first incubated, and later-in order to remove all unbound components, the wells were thoroughly washed. Next, a substrate solution was added to every well. The enzyme (HRP) and substrate were allowed to react over a short incubation period. Only those wells containing MDA and enzyme-conjugated antibody exhibited a colour change. By adding a sulphuric acid solution, the enzyme-substrate reaction was terminated and the colour change was measured spectrophotometrically at a wavelength of 450 nm. A calibration curve was plotted relating the intensity of the colour (O.D) to the concentration of calibrators. The MDA concentration in each sample was interpolated from this calibration curve.

### 2.5. Tissue Analysis

The ovarian and uterine tissue samples were crushed in a liquid nitrogen medium. The powdered, frozen tissue was placed in a test tube containing 500 μL of phosphate-buffered saline (pH = 7.4) previously cooled to 4 °C, and then homogenised using a blade homogeniser (Pro Scientific, PRO200 P/N 01-02200, S/N 02-1167). The homogenates were centrifuged (10,000× *g*, 20 min, 4 °C) and the obtained supernatant was used to assay the activities of antioxidant enzymes, malondialdehyde concentrations and total protein.

Protein concentration in supernatants was determined by the Bradford method, with bovine serum albumin as the standard (Sigma Aldrich, St. Louis, MO, USA, Cat. No. B6916).

Tissue glutathione peroxidase analysis: GPx was analysed using an ELISA kit (Shanghai Sunred Biological Technology Co., Ltd., Shanghai, China, Cat. No. 201-11-1705). A sandwich enzyme immunoassay method was used with this kit. A specific antibody was used to coat the microplate kit that is specific to GPx.

Tissue superoxide dismutase analysis: SOD was analyzed using an ELISA kit (Shanghai Sunred Biological Technology Co., Ltd.; Cat. No. 201-11-0169). A sandwich enzyme immunoassay method was used with this kit. A specific antibody was used to coat the microplate kit that is specific to SOD.

Tissue catalase analysis: CAT was analyzed using an ELISA kit (Shanghai Sunred Biological Technology Co., Ltd.; Cat. No. 201-11-5106). A sandwich enzyme immunoassay method was used with this kit. A specific antibody was used to coat the microplate kit that is exclusive to CAT.

Tissue malondialdehyde analysis: MDA was analyzed using an ELISA kit (Shanghai Sunred Biological Technology Co., Ltd., Cat. No. 201-11-0157). A sandwich enzyme immunoassay method was used with this kit. A specific antibody was used to coat the microplate kit that is exclusive to MDA.

### 2.6. Statistical Analysis

All data are expressed as means ± S.E.M. Differences between groups were analyzed using one-way ANOVA and the Tukey test using Statistica 12.0^®^ program (Statsoft, Tulsa, OK, USA). To assess the homogeneity of variances and normality of distribution, the Levene test and a modified Shapiro–Wilk test were used, respectively. If these assumptions were not met, a logarithmic transformation was applied to the data before ANOVA. Differences between groups were considered significant when *p* ≤ 0.05.

## 3. Results

Table 1 contains data on the amount of consumed feed and selected minerals present in the antioxidant enzymes being assayed. An analysis of the effect of the factors used on the amount of feed intake and that of selected minerals showed no differences between groups in feed intake, while the intake of selected minerals differed, which was related to the different content of these components in individual feeds. The highest intake of all the analyzed minerals was observed in control group animals (CG), while animals from the groups fed modified feed (SG) or periodically starved (PSG) consumed them statistically significantly less.

When analysing the weight of selected organs, the effect of diet composition on the ovarian was found, both in absolute values and per 100 g of body weight—Table 2. A smaller weight of this organ was found in SG and PSG animals compared to CG ones 51.4 ± 5.80 mg and 57.5 ± 8.06 mg vs. 68.1 ± 7.01 mg). There was no effect of the factors used on the uterine weight of the examined female rats.

Analysis of the activity of antioxidant defence enzymes in erythrocytes and malonyldialdehyde concentrations in blood plasma of the examined animals is presented in Figure 1. In the erythrocytes of SG and PSG animals, statistically significantly higher glutathione peroxidase and catalase activities were found compared to CG ones. The activity of superoxide dismutase in erythrocytes and the concentration of MDA in blood plasma were also differed, being significantly higher in PSG animals compared to CG and SG ones.

The results of analyses of the antioxidant defence enzyme activities and malonyldialdehyde concentrations in ovarian homogenates are shown in Figure 2. In the ovarian homogenates of SG animals, the concentration of superoxide dismutase was lower compared to CG ones (8.96 ± 0.40 μg/g protein vs. 10.1 ± 1.21 μg/g protein). Glutathione peroxidase and catalase concentrations were statistically significantly lower in PSG animals compared to CG and SG ones. A different concentration of malonyldialdehyde was also found. Its lowest concentration was observed in the ovarian homogenates from PSG animals (4.31 ± 0.45 μM/g protein), while statistically significantly higher MDA concentration was determined in the ovarian homogenates from CG and SG animals (5.60 ± 0.78 μM/g protein and 5.44 ± 0.60 μM/g protein).

The results of the analyses of the effect of the factors used on the redox balance in uterine homogenates are presented in Figure 3. In the uterine homogenates from SG and PSG animals, statistically significantly higher superoxide dismutase activity was found compared to CG ones (7.11 ± 0.38 μg/g protein and 7.10 ± 0.33 μg/g protein vs. 5.81 ± 0.84 μg/g protein). Glutathione peroxidase and catalase activities, as well as malonyldialdehyde concentrations, were statistically significantly higher in PSG animals compared to CG and SG ones.

Taking into account the determined glucose concentration, it was found that it was statistically significantly higher in SG and PSG animals than in CG ones (5.57 ± 0.61 mmol/l and 5.57 ± 0.38 mmol/l vs. 4.66 ± 0.53 mmol/l)—Table 3. Changes in glucose concentrations were not related to the change in insulin concentration, which was comparable in the examined groups of animals. Statistically significant differences were found in the HOMA-IR index, the value of which was higher in SG and PSG animals compared to CG ones (0.68 ± 0.08 and 0.73 ± 0.13 vs. 0.51 ± 0.11).

When analyzing the concentrations of blood lipid parameters, no effect of the factors used on the concentration of triglycerides was found, which was comparable between the groups. However, the total cholesterol level was statistically significantly lower in PSG animals compared to CG and SG ones (1.71 ± 0.19 mmol/l vs. 2.03 ± 0.16 mmol/l and 2.16 ± 0.19 mmol/l). This was associated with a lower HDL-cholesterol concentration. The LDL-cholesterol concentration was statistically significantly higher in SG and PSG animals compared to CG ones (0.82 ± 0.12 mmol/l and 0.75 ± 0.08 mmol/l vs. 0.58 ± 0.03 mmol/l).

The effect of the applied factors on the concentration of oestrogens in the blood serum of the examined female rats was demonstrated. Their higher concentrations were found in the groups fed with modified feed compared to that fed with control feed; in PSG animals, this concentration was statistically significantly higher not only compared to CG animals, but also to SG ones.

Weight gains recorded during the experiment were statistically significantly higher in SG animals compared to CG and PSG ones (28.3 ± 2.00 g vs. 22.75 ± 2.74 g and 22.5 ± 3.13 g)—Table 4. However, they were not related to the deposition of visceral fat, because its highest amount was found in PSG animals, which deposited it statistically significantly less more compared to CG and SG animals, both in absolute values and per 100 g body weight (3.61 ± 0.65 g/100 g b.w. vs. 2.99 ± 0.37 g/100 g b.w. and 2.77 ± 0.29 g/100 g b.w.).

## 4. Discussion

The results of the conducted research show the relationship between the presence of sucrose in diet and regular periodic starvation when following the sucrose diet and the redox balance in the blood and reproductive organs of female rats.

In the erythrocytes of animals receiving sucrose in their diet amounting to 18% of its energy value (SG animals), an increase in antioxidant defence was observed, manifested by an increase in GPx and CAT activities, whereas no increase in MDA concentration was found. The introduction of one-day starvation per week during the sucrose diet (PSG animals) resulted in increased SOD activity, with simultaneously increased MDA concentration. The increase in the activity of antioxidant enzymes in both groups was observed even with significantly lower intake of components involved in their synthesis (Zn, Mn, Cu, Se, Fe) by animals of these groups.

The increase in ROS production and the intensification of antioxidant processes in the blood and liver of rats, induced by the dietary sugars, were also reported by Maciejczyk [19,20]. The changes observed in our study may have been caused by a higher concentration of glucose found in the blood plasma of SG and PSG animals. Hyperglycaemia induces a number of changes that have a major effect on disturbances in cell metabolism, including redox imbalance, among others by increasing the oxidative processes in mitochondria, increasing the activity of NADPH oxidase (EC 1.6.3.1) and intensified non-enzymatic glycation of proteins, lipids and nucleic acids [21,22]. In the plasma of SG and PSG animals, the insulin sensitivity was also reduced being expressed by a significant increase in the HOMA-IR index. As shown by the study by Kopprasch et al. [23], its values are positively correlated with an increase in ROS production and an increase in the total blood antioxidant potential.

The results of our study seem to indicate that the introduction of regular one-day starvation periods into the sucrose diet intensifies oxidative stress and disturbs oxidative homeostasis. In PSG animals, an increase in the MDA concentration was observed, which indicates an increased ROS production, due to insufficient activity of enzymatic antioxidant processes. The prevalence of ROS production over the antioxidant activity induces oxidative stress, which results in changes in the structure of proteins, lipids, carbohydrates and nucleic acids. The oxidative stress found in PSG animals may have resulted from greater accumulation of visceral adipose tissue compared to the CG and SG animals. Visceral adipose tissue is a source of proinflammatory cytokines, such as IL-6, IL-1β, TNF-α, MCP-1, and can generate ROS via inflammatory processes [24]. The HDL-cholesterol concentration, lower than in other groups, could also have an effect. Experimental studies have shown that HDL, taking into account antioxidant properties of ApoA-I and the presence of enzymes such as paraoxonase 1 and platelet-activating factor acetylhydrolase, has the ability to detoxify the oxidised phospholipids produced during lipid peroxidation [25]. The reason for an increase in MDA concentration in the blood and uterus in this group of animals could also be the higher susceptibility of cell membranes to oxidative damage. Sorensen et al. [26] have found that one of the effects of rat starvation is an increase in the amount of polyunsaturated fatty acids in the cell membranes, which are susceptible to oxidation due to their high chemical reactivity. The increase in membrane lipid peroxidation is accompanied by degenerative changes in tissues, mainly in the vascular area, and activation of risk factors for: hypertension, dyslipidaemia and obesity [27]. The effect of starvation on the redox balance in the blood and liver of rats has been demonstrated by Wasselin [28] and Domenicali [29]. In our study, this effect was also demonstrated in relation to the uterus and ovaries, which may have consequences in female fertility disturbances.

Oxidative stress is one of the factors that also increases the risk of cardiovascular diseases, such as atherosclerosis and hypertension. ROS cause oxidative modification of LDL-C, increase the expression of adhesion molecules ICAM-1 and VCAM-1, and promote the remodelling of blood vessels [30]. In our study, the higher activity of antioxidant enzymes in the erythrocytes of examined SG and PSG female rats was accompanied by unfavourable changes in blood lipid parameters—higher LDL-C concentration and lower HDL-C concentrations. This profile of changes in blood lipid parameters is particularly unfavourable, as it has been shown that low HDL-C levels are an important risk factor for atherosclerotic lesions [31], which was additionally favoured by oxidative stress in the examined animals.

In the uterine homogenates of sucrose-fed animals, an increase in SOD concentration was found. In the uterine homogenates from periodically starved animals, higher GPx and CAT activities and higher MDA concentration were found, whereas in their ovarian homogenates lower GPx and CAT activities and lower MDA concentration compared to CG and SG animals. Disturbances in the redox balance in the uterus and ovaries are therefore significantly higher in PSG animals as compared to SG ones.

In the analyzed tissues, oestrogens could also affect the redox status. Oestrogens play an important role in maintaining the oxidative balance in tissues, among others, by regulating the ROS level in mitochondria. Antioxidant mechanisms of oestrogen activity are also manifested by the increase in SOD activity [32]. Higher activity of SOD was observed in SG animals in the uterus, however, this effect was not found in the ovaries of the animals of the group in which SOD activity was lower compared to animals from the control group. Antioxidant effects of oestrogens depend, among others, on the quantitative ratio of oestrogen receptors ERα:ERβ in cells. It has been found that tissues in which the predominant form is ERβ are characterised by lower susceptibility to oxidative stress [33,34]. Considering the tissue specificity, it has been found that ERβ receptors dominate in the ovaries, prostate, thyroid glands, skin, lungs and bones, whereas ERα receptors dominate in the uterus, testes, liver and kidney [35]. Antioxidant mechanisms of oestrogen activity and the prevalence of ERβ receptors in the ovaries may explain the absence of oxidative stress in the ovaries of SG and PSG animals. In the uterus, an increase in the activity of antioxidant enzymes in SG animals (increase in SOD concentration) and PSG (increase in GPx and CAT concentrations) was observed, probably due to an unfavourable quantitative ratio of ERα: ERβ receptors. Larger changes in the redox status in PSG animals, also manifested by an increase in MDA concentration in the blood and uterus, may paradoxically result from a higher concentration of oestrogens in the plasma of animals of this group. The antioxidant activity of oestrogens depends both on the severity of the factor causing oxidative stress and the concentration of oestrogens. It has been shown that the increase in their concentration, above physiological levels, intensifies ROS generation and leads to oxidative stress [36].

It has been shown that excessive body weight is correlated with higher oestrogen concentration [37]. Perhaps, a higher oestrogen concentration observed in SG animals could have resulted from the higher body weight of examined animals. Obesity and an increase in oestrogen concentration correlated with it may also be the cause of anovulation [38]. This is due to the release of excessive amounts of oestrogen, which inhibits FSH release in the middle of the cycle, blocking ovulation [39]. Obesity disturbs the fertility process not only by increasing oestrogen levels, but also as a consequence of increased insulin level and insulin resistance, which is manifested by an increase in the HOMA-IR index. In PSG animals, an increase in visceral adipose tissue was also observed, which may also indicate an increased proinflammatory activity of adipose tissue as a result of increased cytokine production. Maternal obesity may also have long-term consequences in the deteriorated reproductive state of their offspring. Cheong et al. [40] has observed that diet-induced maternal obesity changes ovarian morphology and gene expression in adult mice offspring. Maternal obesity during pregnancy has long-term harmful consequences for the development of ovaries and the growth of follicles of adult offspring, which may affect their reproductive potential.

The results of this study indicate that the uterus and ovary differ significantly in the oxidation defence status and the amount of ROS generated. In the uterus of PSG animals, an increase in the activity of antioxidant defence enzymes was found, as well as oxidative damage of lipids, confirmed by an increase in the concentration of MDA. In contrast, in the ovaries of this group of animals, the activity of antioxidant enzymes was lower, and ROS generation was also lower, which is indicated by the lower concentration of MDA in the ovaries of starved animals compared to other groups.

Differences in the oxidative status between the uterus and the ovaries also have been reported by Farombi et al. [41]. Using an antimalarial drug (artemisinin), they showed that it caused oxidative damage of lipids in the uterus, manifested by an increase in the concentration of MDA. This effect however, was not found in the ovaries of female rats. Ovaries seem to be protected primarily against the formation of free radicals. In the study by Farombi et al. [41] the concentration of H_2_O_2_, as a marker of the intensity of free radicals reactions, did not change in the ovaries, but increased significantly in the uterus.

It is important to maintain a balance between the amount of ROS and the antioxidant potential to preserve normal tissue function. The appropriate level of ROS is necessary in ovarian cells [9,42], because ROS are involved in the growth of ovarian follicles, mainly by regulating angiogenesis [43]; they also participate in oocyte maturation [44] and ovulation [7]. They also enable hormonal balance, among others by inhibiting progesterone synthesis at the end of the luteal phase of the cycle [45]. In contrast, oxidative stress in the uterus may be one of the causes of recurrent failure of embryo implantation and repeated miscarriages [41].

## 5. Conclusions

In conclusion, it was found that a diet with 16% sucrose contributed to the increase of antioxidant enzyme activity in the blood (GPx and CAT) and uterus (SOD), without changes in MDA concentrations, which indicates an increase in ROS concentration in these tissues, being balanced by an increase in antioxidant enzyme activity. The introduction of a regular one-day starvation period into such a diet intensified oxidative stress and led to a redox imbalance in the reproductive tissues of female rats. This was manifested by higher GPx activity, lower CAT activity and higher MDA concentration in the uterus and lower GPx and CAT activities and lower MDA concentration in the ovaries. The observed changes may have been due to metabolic disorders (greater amount of visceral adipose tissue, higher glucose concentration and HOMA-IR index and lower HDL-C level) and hormonal disorders (higher oestrogen concentration). The observed changes may be the cause of fertility disorders and possible problems with fertilised egg cell implantation into the uterine tissue. The reason being that the reproductive system uses ROS in the processes necessary for reproduction.

## Figures and Tables

**Figure 1 nutrients-11-01544-f001:**
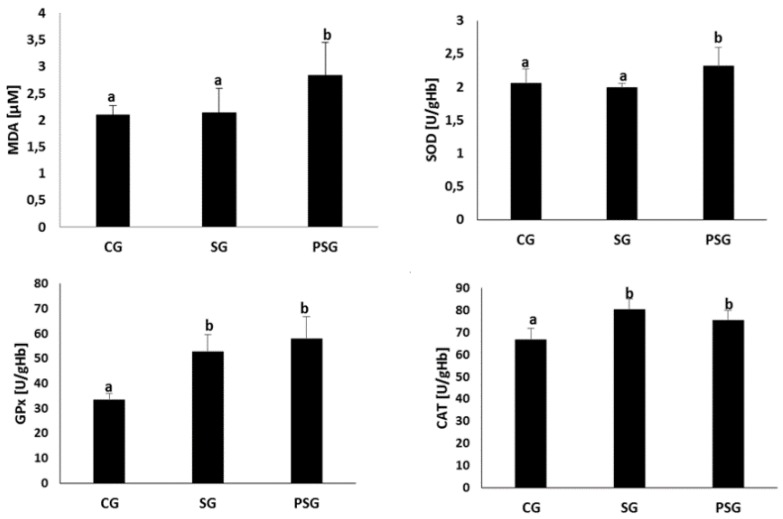
Effect of frequent changes in diet or periodic starvation on plasma malonyldialdehyde (MDA) concentrations and erythrocyte superoxide dismutase (SOD), glutathione peroxidase (GPx), catalase (CAT) activities. CG—Control group, SG—Sugar group, PSG—Periodically starved group, a,b—Means marked with different letters in the same line are statistically different, *p* ≤ 0.05.

**Figure 2 nutrients-11-01544-f002:**
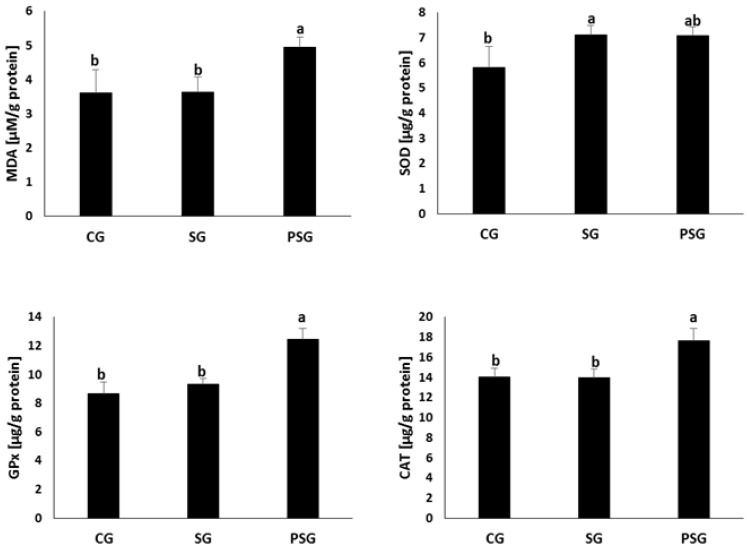
Effect of frequent changes in diet or periodic starvation on ovarian malonyldialdehyde (MDA) concentrations and superoxide dismutase (SOD), glutathione peroxidase (GPx), catalase (CAT) activities. CG—Control group, SG—Sugar group, PSG—Eriodically starved group, a,b—Means marked with different letters in the same line are statistically different, *p* ≤ 0.05.

**Figure 3 nutrients-11-01544-f003:**
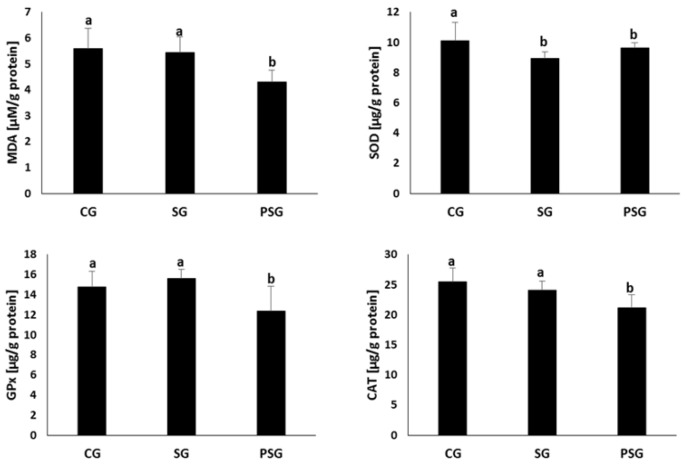
Effect of frequent changes in diet or periodic starvation on uterine malonyldialdehyde (MDA) concentrations and superoxide dismutase (SOD), glutathione peroxidase (GPx), catalase (CAT) activities. CG—Control group, SG—Sugar group, PSG—Periodically starved group, a, b—Means marked with different letters in the same line are statistically different, *p* ≤ 0.05.

**Table 1 nutrients-11-01544-t001:** Effect of frequent changes in diet or periodic starvation on feed, zinc, copper, selenium, iron and manganese intake in the examined rats.

Intake	CG	SG	PSG
Feed intake (g/100 b.w./8 weeks)	335 ± 8.5	326 ± 8.3	333 ± 10.2
Zinc intake (mg/100 b.w./8 weeks)	30.9 ± 0.77 ^b^	25.3 ± 0.76 ^a^	25.8 ± 0.91 ^a^
Copper intake (mg/100 b.w./8 weeks)	6.03 ± 0.15 ^b^	4.58 ± 0.13 ^a^	4.67 ± 0.19 ^a^
Selenium intake (mg/100 b.w./8 weeks)	0.125 ± 0.003 ^b^	0.109 ± 0.002 ^a^	0.111 ± 0.002 ^a^
Iron intake (mg/100 b.w./8 weeks)	67.3 ± 1.68 ^b^	51.5 ± 1.44 ^a^	52.6 ± 2.61 ^a^
Manganese intake (mg/100 b.w./8 weeks)	26.7 ± 0.67 ^b^	21.7 ± 0.63 ^a^	22.2 ± 0.87 ^a^

CG—Control group, SG—Sugar group, PSG—Periodically starved group, ^a,b^—Means marked with different letters in the same line are statistically different, *p* ≤ 0.05.

**Table 2 nutrients-11-01544-t002:** Effect of frequent changes in diet or periodic starvation on rat ovary and uterus weights.

Organ Weight	CG	SG	PSG
Ovary (mg)	68.1 ± 7.01 ^b^	51.4 ± 5.80 ^a^	57.5 ± 8.0 ^a^
(mg/100 g b.w.)	29.6 ± 4.54 ^b^	23.0 ± 2.99	24.9 ± 2.98
Uterus (mg)	483 ± 26.2 ^a^	522 ± 70.7 ^a^	471 ± 67.6 ^a^
(mg/100 g b.w.)	207 ± 20.4 ^a^	226 ± 33.5 ^a^	217 ± 21.7 ^a^

CG—Control group, SG—Sugar group, PSG—Periodically starved group, b.w.—Body weight, ^a,b^—Means marked with different letters in the same line are statistically different, *p* ≤ 0.05.

**Table 3 nutrients-11-01544-t003:** Effect of frequent changes in diet or periodic starvation on glucose, insulin, lipid concentrations and HOMA-IR value in the examined rats.

Traits	CG	SG	PSG
Glucose (mmol/l)	4.56 ± 0.54 ^a^	5.57 ± 0.61 ^b^	5.57 ± 0.38 ^b^
Insulin (mU/mL)	2.51 ± 0.89 ^a^	2.78 ± 0.55 ^a^	2.93 ± 0.87 ^a^
HOMA-IR	0.51 ± 0.11 ^a^	0.68 ± 0.08 ^b^	0.73 ± 0.13 ^b^
TG (mmol/l)	0.48 ± 0.05 ^a^	0.57 ± 0.08 ^a^	0.52 ± 0.10 ^a^
TC (mmol/l)	2.03 ± 0.16 ^b^	2.16 ± 0.19 ^b^	1.71 ± 0.19 ^a^
HDL-C (mmol/l)	1.21 ± 0.08 ^c^	0.94 ± 0.09 ^b^	0.68 ± 0.08 ^a^
LDL-C (mmol/l)	0.58 ± 0.03 ^a^	0.82 ± 0.12 ^b^	0.75 ± 0.08 ^b^
Oestrogens (pg/mL)	14.3 ± 1.84 ^a^	21.2 ± 1.65 ^b^	24.8 ± 3.52 ^c^

TG—Triglycerides, TC—Total cholesterol, HDL-C—HDL-cholesterol, LDL-C—LDL-cholesterol, CG—Control group, SG—Sugar group, PSG—Periodically starved group, ^a,b,c^—Means marked with different letters in the same line are statistically different, *p* ≤ 0.05.

**Table 4 nutrients-11-01544-t004:** Effect of frequent changes in diet or periodic starvation on rat body weight gain and visceral fat.

Traits	CG	SG	PSG
Body weight gain (g)	22.75 ± 2.74 ^a^	28.3 ± 2.00 ^b^	22.5 ± 3.13 ^a^
Visceral fat (g)	6.62 ± 0.48 ^a^	6.24 ± 0.62 ^a^	7.11 ± 0.71 ^b^
(g/100 g b.w.)	2.99 ± 0.37 ^a^	2.77 ± 0.29 ^a^	3.61 ± 0.65 ^b^

CG—Control group, SG—Sugar group, PSG—Periodically starved group, ^a,b^—Means marked with different letters in the same line are statistically different, *p* ≤ 0.05.

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
