# Peer review of "The Impact of a Diet Containing Sucrose and Systematically Repeated Starvation on the Oxidative Status of the Uterus and Ovary of Rats"

_nutrients, 2019, doi:10.3390/nu11071544_

Round 1

Reviewer 1 Report

The manuscript entitled “The impact of a diet containing sucrose and systematically repeated starvation on the oxidative status of the uterus and ovary of rats” by Sadowska et al. is an interesting and comprehensive review. Nevertheless the Authors did not escape some ambiguities in the text. Suggestions of the reviewer are presented below:

1.               The list of all Abbreviation should be added.

2.               The Introduction should explain  why Author choose the 16% administration sucrose diet.

3.               Throughout the body text I found some language and stylish errors. All ambiguities should be corrected.

4.               Results, Table 1S – results should be presented according to the following scheme: 20.1±0,71  (do not use “,”).

5.               Materials and method- the fragment about biochemical analysis should be improved. If it possible I suggests add a table with codes and supplier companies

6.               The reviewer has made some mistakes about references cited in the manuscript. Please unification the reference. The some of them are older than ten years,  and the two (10 and 13) of them have been published even in 1990s. Are there any newer publications concerning this subject? If it is possible, I suggests to change old publications to newer ones.

In spite of the mentioned above small inadequacies, the reviewer thinks that the article after corrections can be published.

Author Response

Dear Reviewer,

We would like to thank you for your time and your review, as well as your remarks, that will undoubtedly make our article even better. We took all your comments into consideration and we have made the following corrections:

According to your suggestion we added a list of abbreviations used in the manuscript.

In the introduction, according to the Reviewer’s suggestion, we explained why we used a diet with 16 % of sucrose composition. (P 2; L 74-75).

Using the help of native speakers we corrected the language and stylish errors.

Thank you for the indicated errors. We corrected the results in Table S1 (currently Table S2).

According to the Reviewer’s suggestion in order to improve the fragment about biochemical analysis in ‘Materials and methods’, we have inserted additions in the text and added Table S3   with codes and supplier companies.

Thank you for your suggestion. We corrected the errors regarding the cited references. We also changed the publication from the 90s to newer articles. However, Reeves et al. [1993] literature must remain in the article.

Yours faithfully,

Joanna Sadowska

Reviewer 2 Report

The reviewer thank the authors for the interesting article. I only have few questions regarding the introduction and method sections.

Line 34: The authors mentioned that "the sugar consumption remains high". Please specify the region(s) and/or the populations.

Line 37: The authors suggested that "total sugars as percentage of energy are highest in infants and young children". Do you mean the total carbohydrate?

Line 54: The reference goes back to this article regarding biochemistry of metal-induced oxidative stress. Please include another reference which is more relevant.

Line 81: Would you please separate the ingredients table from the macronutrients table? And please indicate the energy% of the macronutrients.

Author Response

Dear Reviewer,

We would like to thank you for your time and your review, as well as your remarks, that will undoubtedly make our article even better. We took all your comments into consideration and we have made the following corrections:

According to the suggestion we have completed the literature and specified that we mean developed and developing countries.

Thank you for this remark. "Total sugars" can indeed be understood as "total carbohydrate". That is why we have listed "total sugar" for "sugar" and we added that this applies to all mono- and disaccharides, namely glucose, fructose, lactose, sucrose and maltose.

Thank you for this remark. We changed this publication, currently in the references it is under number 10.

According to the Reviewer Table S1 „Component and chemical composition of diets” we divided into 2 tables Table S1 “Component composition of diets” and Table S2 “Chemical composition of diets”.

Yours faithfully,

Joanna Sadowska